# Keyline in Bean Crop (*Phaseolus vulgaris* L.) for Soil and Water Conservation

**Ma. del Carmen Ponce-Rodríguez [1], Francisco Oscar Carrete-Carreón [2], Gerardo Alonso Núñez-Fernández [3], José de Jesús Muñoz-Ramos [4] and María-Elena Pérez-López [5,*]**

1   Programa Institucional de Doctorado en Ciencias Agropecuarias y Forestales, Universidad Juárez del Estado de Durango, Constitución 404 sur Zona Centro, Durango 34000, Mexico; carmen.ponce@ujed.mx
2   Facultad de Medicina Veterinaria y Zootecnia, Universidad Juárez del Estado de Durango, Carretera Durango-Mezquital, km 11.5, Durango 34307, Mexico; focc1928mx@yahoo.com
3   Maestría en Geomática Aplicada a Recursos Forestales y Ambientales, Universidad Juárez del Estado de Durango, Constitución 404 sur Zona Centro, Durango 34000, Mexico; ganuf.29@gmail.com
4   TecNM, Campus Valle del Guadiana, Carretera Durango-Mexico, km 22.5 s/n, Durango 34371, Mexico; murj59@gmail.com
5   Instituto Politécnico Nacional-Centro Interdisciplinario de Investigación para el Desarrollo Integral Regional Durango, Sigma 119, Fraccionamiento 20 de Noviembre II, Durango 34220, Mexico
*   Correspondence: maelena0359@gmail.com; Tel.: +52-618-158-90-45

**Abstract:** Soil erosion is a common problem worldwide, and in Durango, Mexico, it occurs in 77.4% of territory. Faced with this problem, the hydrological keyline design (HKD) is an alternative that helps to retain soil, increase infiltration, and keep the water uniformly in the land in order to recover its fertility. The objective of this research was to evaluate the effect of HKD on moisture and soil conservation in a rainfed agricultural plot during the spring–summer 2018 cycle with a bean crop (*Phaseolus vulgaris* L.) in the state of Durango, Mexico. Two treatments were established: control and HKD. The variables to measure the effect of the treatments were: soil water content, water erosion, bean yield, and yield components. The results indicated differences ($p < 0.05$) between treatments for the moisture and erosion variables; the HKD retained more water than the control by five percent, while sediment transport was lower in the HKD. No differences ($p > 0.05$) were found regarding bean yield and yield components. However, the yield was 126% higher than regional average in terms of rainfed bean production. Therefore, the implementation of the HKD had a positive impact by retaining soil and moisture.

**Keywords:** erosion; moisture; water; soil; yield

## 1. Introduction

Globally, 33% of the earth's surface shows soil degradation [1]; two billion hectares suffer some type of deterioration [2]. Mexico is also affected in 76% of its surface; in turn, the state of Durango shows erosion in 77.4% of its territory, which is mainly caused by overgrazing, deforestation, and bad agricultural practices [3]. If, adding to this is the fact that more than 50% of the national territory has a dry and semidry climate cause the agroecosystems in these regions to be fragile [4].

An additional factor that causes soil degradation is the farmers' custom of immoderate tillage. Farmers use machinery excessively with the idea of increasing their land's production as well as using excessive practices and agroinputs without considering the deterioration that this entails. Although these practices could initially bring some benefit, as time goes by and with the physical, chemical, and biological changes that occur, in the end, they result in degradation that is severe and difficult to correct [5,6].

Regarding this, Ardisana [7] reported that food production in South America is obtained at the expense of a large ecological footprint, which is mainly due to agricultural production based on traditional practices.

The government of Mexico, with a commitment to participate in soil conservation, developed strategies for the prevention and control of its degradation [8] by implementing actions such as contour furrowing, contour lines, crop rotation, species association, cover crops, living or dead barriers, minimum tillage, rehabilitation of terraces, level stone barriers, raised beds, and infiltration ditches, among others [2]. Despite this, efforts to stop soil degradation have had a low impact, because according to Bolaños et al. [3], there was a 34% increase in damage only from water erosion in a 13-year period. In response to this situation, it is necessary to carry out agricultural conservation activities that allow economic and social development without damaging natural resources [8].

In this regard, one of the most effective conservation agricultural practices is the hydrological keyline design (HKD), which is an alternative tillage system. It was created in Australia in the 1950s for soil and water conservation. To implement this system, it is necessary to know how water moves on the ground, classifying the space in ridges, gullies, watersheds, and drains [9,10]. Therefore, the project always begins with a topographic survey of the site to obtain contour lines vertically 0.5 m apart [11]. On the obtained plane, the key point, which refers to the place on a slope where there is a sudden change in slope, is identified. From this, a level line, which is called a keyline, extends to both sides. Both the key point and keyline are the basis for developing the hydrology design [9].

The HKD is intended to reduce surface runoff, increase infiltration, and conduct-low velocity water from gullies to ridges, which are managed according to the slope with which the keylines are designed. These practices should be followed as a guide to trace the land's tillage pattern (furrows) in a progressive manner both above and below the keyline [11].

Although the HKD has been applied in different parts of Mexico, there is little information that speaks beyond the empirical aspect. Since this technique has been successfully tested, the objective of this work was to evaluate the effect of HKD in a rainfed agricultural plot with beans (*Phaseolus vulgaris* L.) in the state of Durango, where the erosion, soil moisture, and crop productivity were estimated. Based on the above, the following hypothesis was proposed: the application of the HKD favors the retention of moisture and reduces erosion, which results in better crop development.

## 2. Materials and Methods

### 2.1. Location of Study Area

The experiment was carried out in an agricultural plot located 5 km from the town of San José de Tuitán in the municipality of Nombre de Dios, Durango, Mexico, which is 60 km southeast of the city of Durango. The site is located at the geographic coordinates of 23° 59′ 53.35″ N and 104° 14′ 41.04″ W and at an altitude of 1872 m (Figure 1).

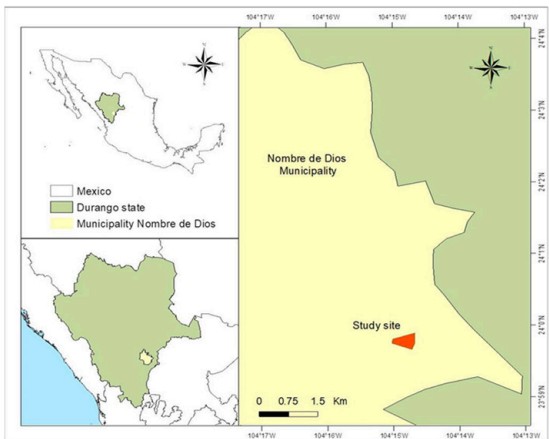

**Figure 1.** Geographic location of the study area in Durango, Mexico.

## 2.2. Site Description

The experimental plot has a total area of 15.36 ha with an average slope of 2.8%, Kastanozem-type soil, and igneous rock [12]. The use of the land is for rainfed agriculture.

## 2.3. Climate

The region's climate is temperate semidry with rains in summer, an average annual rainfall of 450 mm, and an average annual temperature of 17 °C [12]. Additionally, the ambient temperature was recorded every hour from July to December 2019 (Figure 2) by means of a data logger (HOBO®).

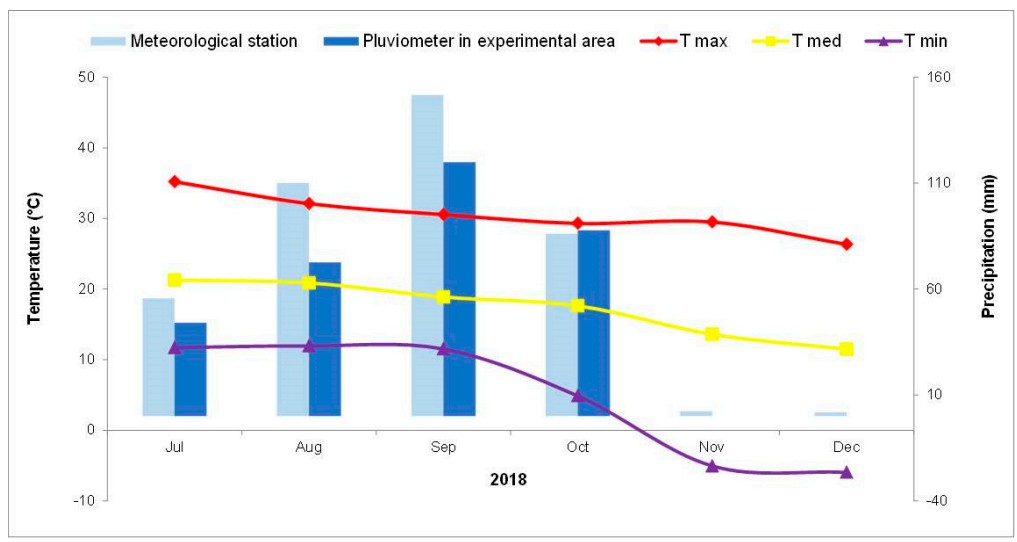

**Figure 2.** Monthly temperature and precipitation data corresponding to the year 2018.

Monthly precipitation was recorded (Figure 2) with the use of a rain gauge fitted with a 20 L drum that was 27 cm in diameter, from which the upper part was cut off. To avoid evaporation, the gauge was assembled inversely on its base so that it was sealed. The monthly precipitation in July, August, September, and October was 44, 73, 120, and 88 mm, respectively.

## 2.4. Soil

### 2.4.1. Chemical Characteristics

Soil samples were taken at four points in the plot at three depths (0–30, 30–60, and 60–90 cm). Not all sampling points reached a depth of 90 cm, since there were superficial calcareous deposits.

The pH, electrical conductivity, and organic matter were determined for the samples obtained in accordance with the Official Mexican Standard NOM-021-2000 [13].

In addition, cations were measured with the aid of a chromatograph (Metrohm 883 Basic IC PLUS brand and model 663 autosampler). Likewise, anions were determined with a chromatograph (Thermo Scientific ICS 1100 brand with As-ap autosampler). The analyses were carried out in the Instituto Politécnico Nacional, Laboratorio de Ciencias Ambientales del Centro Interdisciplinario de Investigación para el Desarrollo Integral Regional, Durango Unit.

### 2.4.2. Physical Characteristics

Twelve sampling points were considered to determine the slope, depth, and texture. According to the texture obtained, the field capacity (FC) and the permanent wilting point

(PWP) were estimated by means of the Bodman and Mahmud equation, which was cited by Silva et al. [14].

$$FC\ (\%\ weight) = 0.023\ (\%\ sand) + 0.25\ (\%\ loam) + 0.61\ (\%\ clay) \qquad (1)$$

$$PWP\ (\%\ weight) = -5 + 0.74\ CC\ \% \qquad (2)$$

The percentage of available water (AW) was obtained from the difference between the FC and the PWP.

### 2.5. Hydrological Design

To implement the HKD, a survey of the plot was first carried out to identify how it received water from both runoff and precipitation (Figure 3a,b) and thereby distribute the water throughout the land and try to meet the crop's needs. The design was applied to the plot's total area, but only 7.0 ha were used for the experimental area (Figure 3c).

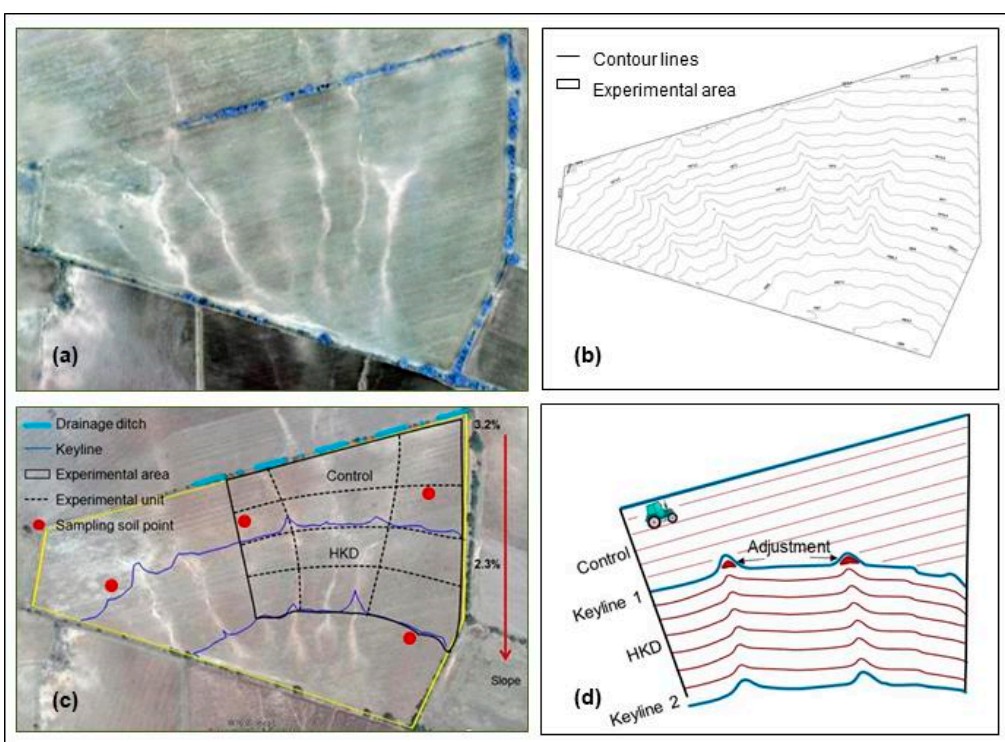

**Figure 3.** Experimental plot: (**a**) Drone image showing the plot's initial condition; (**b**) topographic plane of level curves of the experimental area; (**c**) location of keyline and treatments; (**d**) sketch of the furrow pattern.

In the upper part of the plot, a channel was created to prevent the entry of runoff water, so that the experimental area received exclusively rainwater (Figure 3c).

To obtain the contour map, a grid of points equidistant at 20 m was created on the ground with the support of a total topographic station and a fixed-wing drone (Figure 3b). Two curves were identified on the plane as keylines in accordance with the Gras [9] methodology. Keyline 1 was established as a level channel-ditch to retain runoff and distribute it throughout the experimental area (Figure 3c).

After making some adjustments (Figure 3d), Keyline 1 was used as a guide to design the sowing and tillage pattern, which was drawn progressively below and parallel to Keyline 1 until reaching Keyline 2. When gullies were identified on the ground, the furrow line was smoothly delineated above them, while on the ridges it was traced slightly below. In addition, a section was left without a design to be worked in the traditional way (straight furrow, Figure 3d).

*2.6. Experiment Design*

To know the water retention, erosion, and productivity in the Pinto Saltillo-variety bean crop (*Phaseolus vulgaris* L.), the following treatments were established:

1. Control: traditional furrowing in a straight manner according to the land's upper boundary.
2. HKD: the furrow was made parallel to the keyline.

Soil preparation and farming work were in accordance with the traditional method (fallowing, harrowing, sowing, weeding, fertilization, etc.). Since the rainy season was delayed, bean planting was carried out late, in the first week of August 2018, when it should occur no later than the third week of July. The resulting population density was 70,000 plants per hectare. Fertilization was carried out by applying the foliar fertilizer "Supermagro", which is handmade. There was no need to apply any pesticide, since there were no weeds, pests, or diseases in the crop.

*2.7. Variables Analyzed*

In the soil: moisture, available water, and erosion.
In the vegetation: bean yield, number of pods per plant, and number of beans per pod.
The methodology is described below.

2.7.1. Soil Water Content (%)

Soil water content was measured at the end of the crop cycle by means of time-domain reflectometry. For this, a sensor with automatic recording (Time Domain Reflectometer—TDR, Campbell Scientific) with a 14 cm probe was used. The AV was estimated by the difference between the moisture content and the permanent wilting point.

2.7.2. Erosion (ton ha$^{-1}$)

To quantify erosion, the layers of transported and retained solids were measured by the keyline located downstream of each treatment (Control and HKD, Figure 3d) with the help of a digital vernier (Truper$^®$ brand) to an approximation of hundredths of a millimeter. Measurements were made at 18 points located along each keyline. The results obtained were converted to tons per hectare. The record was taken on September 19 after 150 mm of cumulative precipitation.

2.7.3. Yield (kg ha$^{-1}$)

The bean yield was estimated in each treatment (Figure 3c). In each treatment, four 5 m sampling sites were established, which were experimental units in which all the bean plants were harvested. From these, three plants were taken per experimental unit to determine the yield components, such as number of pods per plant and number of beans per pod.

*2.8. Statistical Analysis*

The data were subjected to a normality analysis using the Shapiro–Wilk test. When the variables met the normality requirement, a variance analysis was carried out. In the cases where there were significant differences, the comparison of means was carried out with the Tukey test ($p < 0.05$). When the data did not show normality, the nonparametric Kruskal–Wallis test was performed. Statistical analyzes were performed with the program STATISTICA version 7 [15].

**3. Results**

*3.1. Soil*

3.1.1. Chemical Characteristics

The results in Table 1 showed that the soil was moderately alkaline [16] and low in organic matter and nutrients.

**Table 1.** Average (*n* = 4) of the soil's physicochemical characteristics.

| Parameters | Depth | | | | |
| --- | --- | --- | --- | --- | --- |
| | 0–30 | 30–60 | 60–90 | Mean | ±SEM Error |
| pH | 8.0 | 7.9 | 8.0 | 7.95 | 0.05 |
| Electric conductivity (dS m$^{-1}$) | 0.27 | 0.26 | 0.25 | 0.26 | 0.005 |
| Organic matter (%) | 1.0 | 1.3 | 1.3 | 1.17 | 0.04 |
| Nitrites (mg kg$^{-1}$) | 1.5 | 1.7 | 1.3 | 1.5 | 0.08 |
| Nitrates (mg kg$^{-1}$) | 9.3 | 5.9 | 7.3 | 8.0 | 0.86 |
| NH$_3$/NH$_4^{+1}$ (mg kg$^{-1}$) | 13 | 9 | 9 | 11.1 | 1.54 |
| Phosphates (mg kg$^{-1}$) | NA | NA | NA | NA | NA |
| K$^{+1}$ (mg kg$^{-1}$) | 17 | 11 | 9 | 14 | 1.44 |
| Na$^{+1}$ (mg kg$^{-1}$) | 20 | 22 | 15 | 19 | 2.67 |
| Ca$^{+2}$ (mg kg$^{-1}$) | 57 | 58 | 57 | 57 | 1.89 |
| Mg$^{+2}$ (mg kg$^{-1}$) | 3 | 2 | 3 | 3 | 0.25 |
| Chlorides (mg kg$^{-1}$) | 20 | 32 | 20 | 23 | 3.95 |
| Sulfates (mg kg$^{-1}$) | 15 | 14 | 36 | 20 | 3.86 |

SEM = standard error of the mean; NA = not available.

### 3.1.2. Physical Characteristics

As shown in Figure 4, the work site had shallow soil with the texture of clay loam, with 43 cm in the upper part of the plot and 71 cm in the lower part.

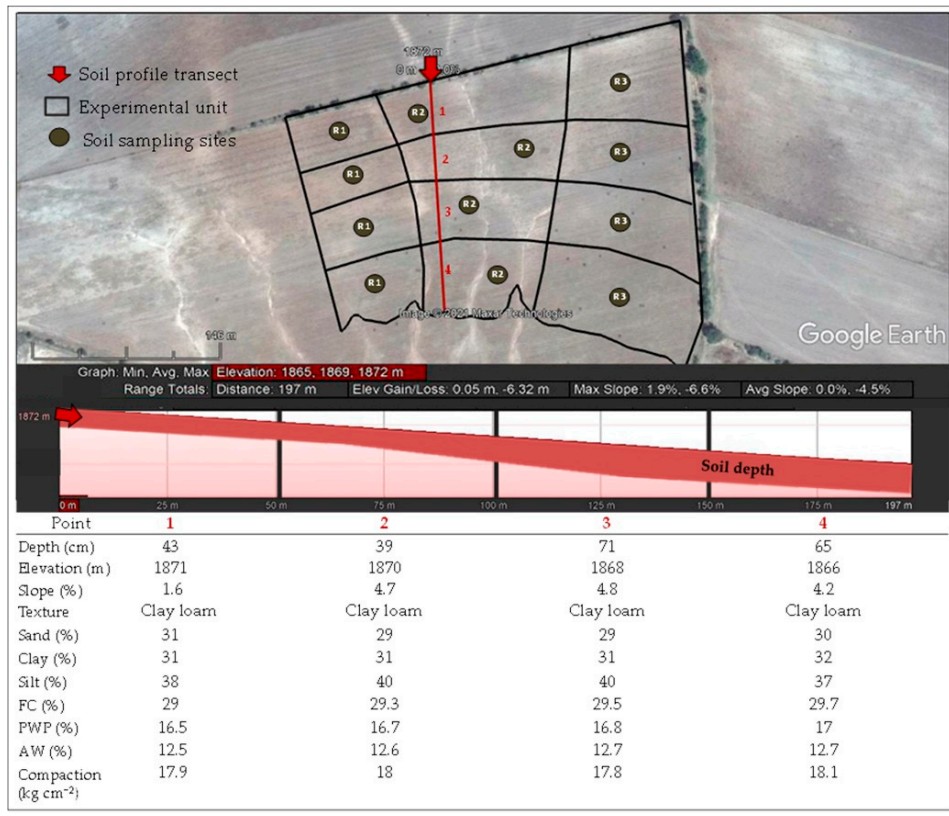

**Figure 4.** Longitudinal profile of the terrain studied and averages for the physical characteristic (*n* = 3).

Because of the soil's texture, it had slow permeability at 0.25 cm hr$^{-1}$, compaction of 18 kg cm$^{-2}$, and average field capacity of 29% with a permanent wiling point of 16% and average available water of 13% for all four points [16].

### 3.2. Moisture (%)

The moisture content in the first 14 cm of the soil after the rainy season showed differences ($p < 0.05$) between treatments at 32% and 27% for the areas with HKD and control, respectively (Figure 5).

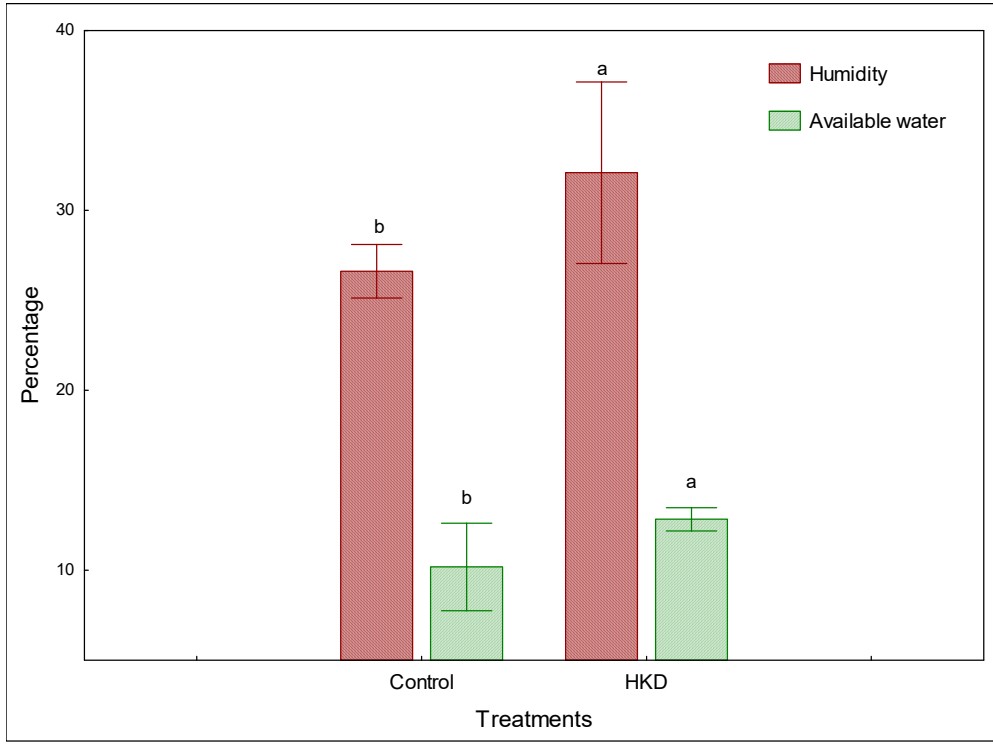

**Figure 5.** Averages of the moisture percentage and water available in the soil surface layer ($n = 6$).

Similarly, the available water showed statistically significant differences ($p < 0.05$) between the treatments, where the HKD had 13% surpassed the control that had a 10%.

### 3.3. Erosion (ton ha$^{-1}$)

Significant differences ($p < 0.05$, Figure 6) in erosion were observed between treatments. The HKD resulted in less transported solids ($86 \pm 23$ ton ha$^{-1}$) than the control ($115 \pm 23$ ton ha$^{-1}$).

The statistical difference observed between the treatments indicates that the tillage pattern established in the HKD (Figure 3d) reduced runoff and thus soil movement. According to Montes et al. [17], the classification rank of water erosion was medium in the HKD but considerable in the control area.

### 3.4. Vegetation

There were no statistical differences ($p > 0.05$) between the control and the HKD for: bean yield (the average of which was 654 kg ha$^{-1}$); the number of pods per plant (15 pods) and for the number of grains per pod (6 per pod). However, it is important to note that this performance was 33.5% higher than the national average and 125% higher than the regional average for 2018.

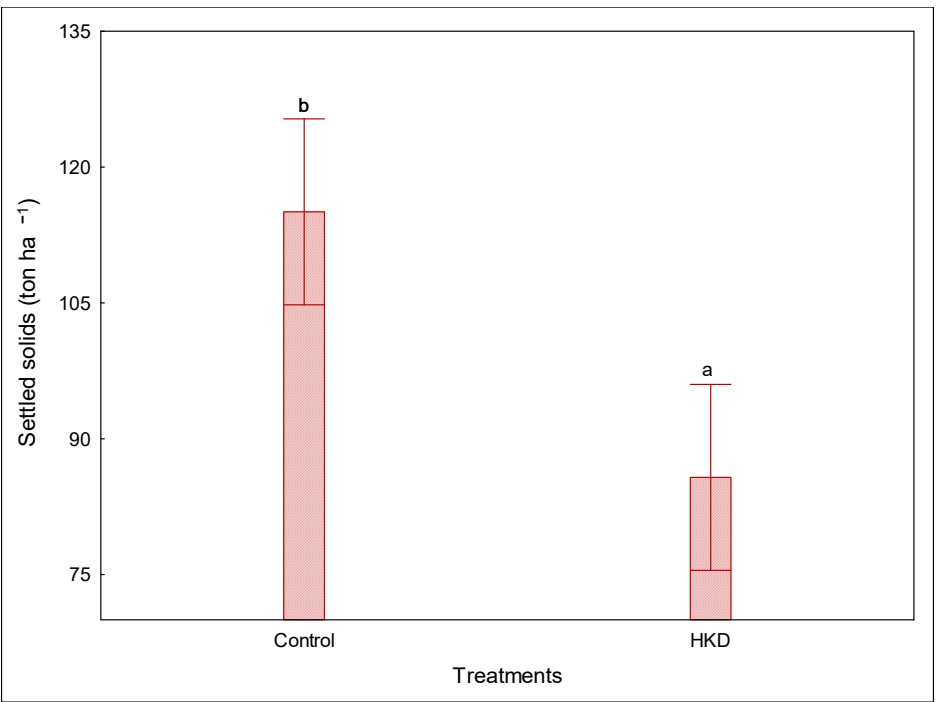

**Figure 6.** Average of soil eroded in each treatment (*n* = 18).

### 3.5. Graphic—Visual Effect of the Evaluated Treatments

　　Figure 7a shows how water flow, due to the slope of the furrows, in the control area and how the keyline stopped the runoff and the distributed it. Photo in the Figure 7b shows the HKD fulfilling its function of distributed and the retaining water in the furrow after a rain event.

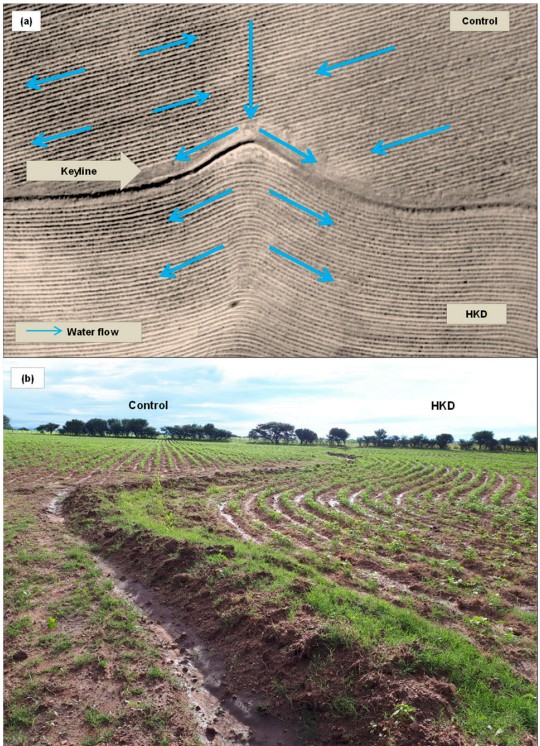

**Figure 7.** Experimental area with bean cultivation established in both treatments: (**a**) Water flow according to the tillage pattern [18]; (**b**) HKD retaining water in furrows after a storm.

## 4. Discussion

According to topographic survey, diagnosis, and characterization, the work site has a catchment area of approximately 96 ha, which for decades has concentrated and directed runoff over gently the terrain. This, in addition to the furrow that has been used for years and that favors the slope, has caused soil and fertility loss and the formation of gullies by erosion. The soils have shallow depth, with 50 cm in the upper part and up to 71 cm in the lower part. The lack of soil in the upper part decreases its capacity to retain water. This is avoided by handling the terrain with HKD [5,16].

Based on this and following the methodology to establish the HKD as applied by Gras [9], the key point (which marks the site on the slopes where the water is concentrated) was determined in the experimental area. Starting from here, the line that would become the keyline was determined. Once traced on the ground and followed in the tillage with some adjustments, it would allow infiltration to be favored as opposed to runoff by slowing down and distributing water throughout the space, with positive results in water and soil conservation.

Regarding soil moisture, the results indicate that the hydrological design established in the area influenced the water to remain in the soil longer. This is in line with what was described by Ponce et al. [19], who found higher moisture content in silvopastoral soils where this design was applied.

Some authors have reported that the presence of established conservation works based on contour lines cuts the hydrological connectivity of the slope, which slows surface runoff, extends the infiltration time throughout its structure, and lets moisture remain on the ground for a longer term [20,21].

Figure 7 explains this phenomenon. It shows how the cultivation pattern in the control area leads the water flow to the streams, causing it to quickly leave the ground. However, in the area with the HKD, the tillage pattern with the keyline stops the flow of water coming from the slope and leads and distributes it over the infrastructure of furrows as described by Gras [9].

Khokhar et al. [22], found that different slopes affect the yield, when intercropped between corn and copui, where, the higher the slope, the greater the loss of soil, so if the HKD is used, these slopes would not affect the overall yield. For their part, Jadhav et al. [23] implemented HKD to establish a green barrier crop with the legume Leucaena (Subabool) and found that the system reduced runoff and soil erosion and increased moisture content in cultivated fields.

Other conservation mechanisms such as the use of terraces vary from HKD in that the former makes the water move level on the ground but is established without any slope. In contrast, slopes are created in HKD that allow changing the speed and distribution of water over the plot as needed [9,24].

HKD, like conservation tillage methods such as permanent planting basins, proved to be more beneficial than conventional methods for degraded soils. By reducing erosion, it conserves the soil and thereby improves its fertility with the short- and medium-term benefit of increasing the land's productivity, which leads to better harvests [10,25,26].

The results in the experimental area confirmed these facts. The decrease in settled solids indicates that the established tillage pattern (Figure 3d) reduced runoff and, therefore, soil movement. The HKD made it possible to lower the speed of the water on the ground and thus reduce erosion, avoiding the loss of 29 tons ha$^{-1}$ in the study area. When erosion is contained, it improves the hydraulic characteristics of the soil by increasing its thickness. This increases productivity and then improves the content of organic matter, which, according to the FAO [27], leads to an increase in the infiltration rate since organic matter favors the recovery of surface porosity, which in this work was not measured. However, the field capacity, point of permanent wilting, and AW were determined with values of 29, 17, and 13% respectively. The HKD treatment showed the highest percentage of available water. Although, in the control area, hydrological connectivity is reduced by the effect of the perpendicularity of the furrows, not respecting the contour curves

when making them straight, generates gullies or streams when heavy rains occur that also dragged solids in their path, according to explained Gras, Figure 3a [9].

Optimizing resources in rainfed areas with HKD contributes to lowering the ecological footprint because it makes the land more productive with less energy and inputs. In this vein, the bean yield obtained in the experimental plot was 126% higher than the national average production in the region (290 kg ha$^{-1}$) [28,29] according to reports from the Servicio de Información Agroalimentaria y Pesquera [30].

It is worth mentioning that the study area belongs to the rainfed zone of the Mexican Altiplano, which is characterized by shallow soils, low moisture retention capacity, and reduced and erratic precipitation, all of which contribute to low yield of crops [31,32].

The yield components for the two treatments surpassed those obtained by other authors such as Domínguez and Bello [33], who found 8.6 pods per plant and 3.25 beans per pod for the same species when fertilized with vermicompost.

The improvement in production indicates that HKD influenced the water to remain in the soil, and that the crop made better use of it by retaining it for a longer time, even though the precipitation was lower than the average (324 mm, being 450 mm the annual average). Klik et al. [34] explained that the infrastructure created retains water for longer periods and extends the crop's growing season even after the rainy season, which allows yields to increase.

The foregoing coincides with the IMTA's [35] research on the subject, which achieved positive results in bean yield, and with Valdez and Aramayo [36] in biomass production when implementing HKD. Therefore, it is expected that in the short- and medium-term, if the design and application of biofertilizers is followed, the bean yield will be higher in the experimental plot.

The HDK did not affect the soil's physicochemical properties, which according to the FAO [37] takes more time to manifest. However, it is expected that in the medium- and long-term, by reducing runoff, the dragging of the particles is retained, increasing the soil layer. The greater the soil layer, the more space there is for water retention and the activation of microbiology; therefore, the site's productivity is expected to increase, as "the available water capacity associated with the depth of the soil determines the volume of water usable by plants in a particular place" [27].

## 5. Conclusions

The implementation of the HKD proved to be an efficient alternative for soil and water conservation in the evaluated area because:

- The infrastructure provided by the HKD changed the terrain's hydrology, distributing the water on the surface in a homogeneous manner and decreasing the transported solids by reducing the speed of runoff.
- The HKD showed positive effects in the bean crop, where the yield obtained exceeded the regional average by 125%.
- The establishment of the HKD together with the use of biofertilizers supports the production of organic food and contributes to lowering the ecological footprint.

**Author Contributions:** Conceptualization, M.-E.P.-L. and M.d.C.P.-R.; methodology, J.d.J.M.-R and M.-E.P.-L.; software, G.A.N.-F. and J.d.J.M.-R.; validation, M.-E.P.-L.; formal analysis, M.d.C.P.-R., J.d.J.M.-R., M.-E.P.-L. and G.A.N.-F; investigation, M.d.C.P.-R. and G.A.N.-F.; resources, M.d.C.P.-R., M.-E.P.-L. and F.O.C.-C.; data curation, F.O.C.-C., J.d.J.M.-R., and G.A.N.-F.; writing—original draft preparation, M.d.C.P.-R.; writing—review and editing, F.O.C.-C. and M.-E.P.-L.; visualization, F.O.C.-C. and M.-E.P.-L.; supervision, M.-E.P.-L. and F.O.C.-C.; project administration M.d.C.P.-R. and M.-E.P.-L.; funding acquisition, M.d.C.P.-R. and M.-E.P.-L. All authors have read and agreed to the published version of the manuscript.

**Funding:** This research received no external funding.

**Institutional Review Board Statement:** Not applicable.

**Informed Consent Statement:** Not applicable.

**Data Availability Statement:** Not applicable.

**Acknowledgments:** The first author thanks CONACYT for the financial support to carry out her doctoral studies; the Programa Institucional de Doctorado en Ciencias Agropecuarias y Forestales for financial aid; and Miguel Ángel Pulgarin Gámiz, by lending his authority to establish this research.

**Conflicts of Interest:** The authors declare no conflict of interest.

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
