# Peer review of "Keyline in Bean Crop (Phaseolus vulgaris L.) for Soil and Water Conservation"

_sustainability, doi:10.3390/su13179982_

Round 1

Reviewer 1 Report

The discussion needs to be improved. I would like to see more discussion on water retention of HDK, as it improved it. Did HDK affect the physical properties of the soil or it was due to the special topography of the region? Didn't you measure the hydraulic properties of the soil? Your discussion is very general. 

Although the national average is important, you should discuss based on the average for the location you worked!

Conclusion should be expanded and preferably stated as bullet points.

Author Response

Dear Reviewer

Thank you very much for the comments and suggestions, with base in them, we separated the information regarding the characteristics of the soil into chemical and physical, where in the latter we added results that we had not included, which we have used to achieve a better description of the site of work.

Observation   

English language and style: English language and style are fine/minor spell check required

Answer

The document was reviewed by a second time by native a English-speaker of Canadian Institute of Modern Languages in Durango, México.

  1. Observation
    1. The discussion needs to be improved. I would like to see more discussion on water retention of HKD, as it improved it.
    2. Did HKD affect the physical properties of the soil, or it was due to the special topography of the region?
    3. Didn't you measure the hydraulic properties of the soil? Your discussion is very general.

Answer

  1. The discussion was improved, we also explain in more detail the development of the key line in the introduction and discussion.
  2. Now, the HKD did not affect the physical properties of the soil, however, it is expected that in the medium and long term it will happen, because by reducing runoff, the drag of soil particles decreases by increasing their volume, where there is more soil, there will be more space to store water and improve the productivity.
  3. Thus, now, the results obtained were mainly due to the management of water according to the special topography of the region, where the speed of runoff in the streams was reduced and was driven to the slopes.
  4. Not all the hydraulic properties of the soil were evaluated, however, it is considered that the revised variables reflect the impact of the implementation of the HKD in the soil. But your observation will be considered in future evaluations.

  1. Observation

Although the national average is important, you should discuss based on the average for the location you worked!

Answer

The drafting of the paragraph was modified as mentioned by the reviewer.

  1. Observation

Conclusion should be expanded and preferably stated as bullet points.

Answer

The author's rules of the journal indicate that it is optional to write conclusions. They explain that it can be added to the manuscript if the discussion is long or complex.

However, the conclusions were drafted with vignettes following the reviewer's suggestions.

Best regards

Authors

Reviewer 2 Report

  1. Lines 28-29: Soybean yield in the experimental plots were greater than the national and regional average. This comparison was run in different scales. I cannot agree that it is a meaningful comparison.
  2.  Line 91: The axis of precipitation in Figure 2 must be revised. There is negative value for precipitation?
  3. Lines 99-101: Soil sample sites should be shown in Figure 3. 

Author Response

Dear reviewer

Thank you very much for your comments and suggestions, which we have addressed, one by one, and are described below.

Observation  

English language and style: Moderate English changes required

Answer

The document was reviewed by a second time by native a English-speaker of Canadian Institute of Modern Languages in Durango, México.

  1. Observation

Lines 28-29: Soybean yield in the experimental plots were greater than the national and regional average. This comparison was run in different scales. I cannot agree that it is a meaningful comparison.

Answer

The bean yield values are reported by the “SIAP” (Agrifood and Fisheries Information Service), which is a decentralized body of the Ministry of Agriculture and Rural Development in Mexico.

The regional values presented correspond to the municipality where the experimental plot is located, and it is itself included within the statistics. Therefore, it was considered as a reliable source as a reference to the performance of the area and later at the national level, in no way the intention is to make a statistical comparison and present significant differences.

However, to address the reviewer's observation, the wording in line 300 was modified.

  1. Observation

Line 91: The axis of precipitation in Figure 2 must be revised. There is negative value for precipitation?

Answer

Figure 2 was examined, the precipitation values (represented in blue bars) do not contain negative values. However, the maximum, median and minimum temperature averages (represented by lines in red, yellow, and purple, respectively), include negative values in the minimum average temperature line.

  1. Observation

Lines 99-101: Soil sample sites should be shown in Figure 3.

Answer

The Figure 3 was fixed as mentioned by the reviewer.

In addition, Figure 4 was added, where the sampling points where the physical characteristics of the soil were determined are shown.

Best regards

Authors

Round 2

Reviewer 1 Report

Accepted